# A PAC-Bayesian Approach to Spectrally-Normalized Margin Bounds for Neural Networks

**Behnam Neyshabur, Srinadh Bhojanapalli, Nathan Srebro**
Toyota Technological Institute at Chicago
{bneyshabur, srinadh, nati}@ttic.edu

## Abstract

We present a generalization bound for feedforward neural networks with ReLU activations in terms of the product of the spectral norm of the layers and the Frobenius norm of the weights. The key ingredient is a bound on the changes in the output of a network with respect to perturbation of its weights, thereby bounding the sharpness of the network. We combine this perturbation bound with the PAC-Bayes analysis to derive the generalization bound.

## 1 Introduction

Learning with deep neural networks has enjoyed great success across a wide variety of tasks. Even though learning neural networks is a hard problem, even for one hidden layer (Blum & Rivest, 1993), simple optimization methods such as stochastic gradient descent (SGD) have been able to minimize the training error. More surprisingly, solutions found this way also have small test error, even though the networks used have more parameters than the number of training samples, and have the capacity to easily fit random labels (Zhang et al., 2017).

Harvey et al. (2017) provided a generalization error bound by showing that the VC dimension of d-layer networks is depth times the number parameters improving over earlier bounds by Bartlett et al. (1999). Such VC bounds which depend on the number of parameters of the networks, cannot explain the generalization behavior in the over parametrized settings, where the number of samples is much smaller than the number of parameters.

For linear classifiers we know that the generalization behavior depends on the norm and margin of the classifier and not on the actual number of parameters. Hence, a generalization bound for neural networks that only depends on the norms of its layers, and not the actual number of parameters, can explain the good generalization behavior of the over parametrized networks.

Bartlett & Mendelson (2002) showed generalization bounds for feedforward networks in terms of unit-wise $\ell_1$ norm with exponential dependence on depth. In a more recent work, Neyshabur et al. (2015a) provided generalization bounds for general class of group norms including Frobenius norm with same exponential dependence on the depth and showed that the exponential dependence is unavoidable in the worst case.

Bartlett et al. (2017a) showed a margin based generalization bound that depends on spectral norm and $\ell_1$ norm of the layers of the networks. They show this bound using a complex covering number argument. This bound does not depend directly on the number of parameters of the network but depends on the norms of its layers. However the $\ell_1$ term has high dependence on the number of hidden units if the weights are dense, which is typically the case for modern deep learning architectures.

Keskar et al. (2016) suggested a sharpness based measure to predict the difference in generalization behavior of networks trained with different batch size SGD. However sharpness is not a scale invariant measure and cannot predict the generalization behavior (Neyshabur et al., 2017). Instead sharpness when combined with the norms of the network can predict the generalization behavior according to the PAC-Bayes framework (McAllester, 1999). Dziugaite & Roy (2017) numerically evaluated a generalization error bound from the PAC-Bayes framework showing that it can predict the difference in generalization behavior of networks trained on true vs random labels. This generalization result is

more of computational in nature and gives much tighter (non-vacuous) bounds compared with the ones in either (Bartlett et al., 2017a) or the ones in this paper.

In this paper we present and prove a margin based generalization bound for feedforward neural networks with ReLU activations, that depends on the product of the spectral norm of the weights in each layer, as well as the Frobenius norm of the weights.

Our generalization bound shares much similarity with a margin based generalization bound recently presented by Bartlett et al. (2017a). Both bounds depend similarly on the product of the spectral norms of each layer, multiplied by a factor that is additive across layers. In addition, Bartlett et al. (2017a) bound depends on the elementwise $\ell_1$-norm of the weights in each layer, while our bound depends on the Frobenius (elementwise $\ell_2$) norm of the weights in each layer, with an additional multiplicative dependence on the "width". The two bounds are thus not directly comparable, and as we discuss in Section 3, each one dominates in a different regime, roughly depending on the sparsity of the weights. We also discuss in what regimes each of these bounds could dominate a VC-bound based on the overall number of weights.

More importantly, our proof technique is entirely different, and arguably simpler, than that of Bartlett et al. (2017a). We derive our bound using PAC-Bayes analysis, and more specifically a generic PAC-Bayes margin bound (Lemma 1). The main ingredient is a perturbation bound (Lemma 2), bounding the changes in the output of a network when the weights are perturbed, thereby its sharpness, in terms of the product of the spectral norm of the layers. This is an entirely different analysis approach from the covering number analysis of Bartlett et al. (2017a). We hope our analysis can give more direct intuition into the different ingredients in the bound and will allow modifying the analysis, e.g. by using different prior and perturbation distributions in the PAC-Bayes bound, to obtain tighter bounds, perhaps with dependence on different layer-wise norms.

We note that other prior bounds in terms of elementwise or unit-wise norms (such as the Frobenius norm and elementwise $\ell_1$ norms of layers), without a spectral norm dependence, all have a multiplicative dependence across layers or exponential dependence on depth (Bartlett & Mendelson, 2002; Neyshabur et al., 2015b), or are for constant depth networks (Bartlett, 1998). Here only the spectral norm is multiplied across layers, and thus if the spectral norms are close to one, the exponential dependence on depth can be avoided.

After the initial preprint of this paper, Bartlett et al. (2017b) presented an improved bound that replaces the dependency on $\ell_1$ norm of the layers (Bartlett et al., 2017a) with $\ell_{2,1}$ norm. This new generalization bound is strictly better than our Frobenius norm bound, and improves over existing results.

## 1.1 Preliminaries

Consider the classification task with input domain $\mathcal{X}_{B,n} = \left\{ \mathbf{x} \in \mathbb{R}^n \mid | \sum_{i=1}^n x_i^2 \leq B^2 \right\}$ and output domain $\mathbb{R}^k$ where the output of the model is a score for each class and the class with the maximum score will be selected as the predicted label. Let $f_{\mathbf{w}}(\mathbf{x}) : \mathcal{X}_{B,n} \to \mathbb{R}^k$ be the function computed by a $d$ layer feed-forward network for the classification task with parameters $\mathbf{w} = \text{vec}\left(\{W_i\}_{i=1}^d\right)$, $f_{\mathbf{w}}(\mathbf{x}) = W_d\, \phi(W_{d-1}\, \phi(....\phi(W_1 \mathbf{x})))$, here $\phi$ is the ReLU activation function. Let $f_{\mathbf{w}}^i(\mathbf{x})$ denote the output of layer $i$ before activation and $h$ be an upper bound on the number of output units in each layer. We can then define fully connected feedforward networks recursively: $f_{\mathbf{w}}^1(\mathbf{x}) = W_1 \mathbf{x}$ and $f_{\mathbf{w}}^i(\mathbf{x}) = W_i \phi(f_{\mathbf{w}}^{i-1}(\mathbf{x}))$. Let $\|.\|_F$, $\|.\|_1$ and $\|.\|_2$ denote the Frobenius norm, the element-wise $\ell_1$ norm and the spectral norm respectively. We further denote the $\ell_p$ norm of a vector by $|.|_p$.

**Margin Loss.** For any distribution $\mathcal{D}$ and margin $\gamma > 0$, we define the expected margin loss as follows:

$$L_\gamma(f_{\mathbf{w}}) = \mathbb{P}_{(\mathbf{x},y)\sim\mathcal{D}} \left[ f_{\mathbf{w}}(\mathbf{x})[y] \leq \gamma + \max_{j \neq y} f_{\mathbf{w}}(\mathbf{x})[j] \right] \tag{1}$$

Let $\widehat{L}_\gamma(f_{\mathbf{w}})$ be the empirical estimate of the above expected margin loss. Since setting $\gamma = 0$ corresponds to the classification loss, we will use $L_0(f_{\mathbf{w}})$ and $\widehat{L}_0(f_{\mathbf{w}})$ to refer to the expected risk and the training error. The loss $L_\gamma$ defined this way is bounded between 0 and 1.

## 1.2 PAC-BAYESIAN FRAMEWORK

The PAC-Bayesian framework (McAllester, 1998; 1999) provides generalization guarantees for randomized predictors, drawn form a learned distribution $Q$ (as opposed to a learned single predictor) that depends on the training data. In particular, let $f_{\mathbf{w}}$ be any predictor (not necessarily a neural network) learned from the training data and parametrized by $\mathbf{w}$. We consider the distribution $Q$ over predictors of the form $f_{\mathbf{w}+\mathbf{u}}$, where $\mathbf{u}$ is a random variable whose distribution may also depend on the training data. Given a "prior" distribution $P$ over the set of predictors that is independent of the training data, the PAC-Bayes theorem states that with probability at least $1 - \delta$ over the draw of the training data, the expected error of $f_{\mathbf{w}+\mathbf{u}}$ can be bounded as follows (McAllester, 2003):

$$\mathbb{E}_{\mathbf{u}}[L_0(f_{\mathbf{w}+\mathbf{u}})] \leq \mathbb{E}_{\mathbf{u}}[\widehat{L}_0(f_{\mathbf{w}+\mathbf{u}})] + 2\sqrt{\frac{2\left(KL\left(\mathbf{w}+\mathbf{u}\|P\right) + \ln\frac{2m}{\delta}\right)}{m-1}}. \tag{2}$$

To get a bound on the expected risk $L_0(f_{\mathbf{w}})$ for a single predictor $f_{\mathbf{w}}$, we need to relate the expected perturbed loss, $\mathbb{E}_{\mathbf{u}}[L_0(f_{\mathbf{w}+\mathbf{u}})]$ in the above equation with $L_0(f_{\mathbf{w}})$. Toward this we use the following lemma that gives a margin-based generalization bound derived from the PAC-Bayesian bound (2):

**Lemma 1.** *Let $f_{\mathbf{w}}(\mathbf{x}) : \mathcal{X} \to \mathbb{R}^k$ be any predictor (not necessarily a neural network) with parameters $\mathbf{w}$, and $P$ be any distribution on the parameters that is independent of the training data. Then, for any $\gamma, \delta > 0$, with probability $\geq 1 - \delta$ over the training set of size $m$, for any $\mathbf{w}$, and any random perturbation $\mathbf{u}$ s.t. $\mathbb{P}_{\mathbf{u}}\left[\max_{\mathbf{x}\in\mathcal{X}} |f_{\mathbf{w}+\mathbf{u}}(\mathbf{x}) - f_{\mathbf{w}}(\mathbf{x})|_{\infty} < \frac{\gamma}{4}\right] \geq \frac{1}{2}$, we have:*

$$L_0(f_{\mathbf{w}}) \leq \widehat{L}_{\gamma}(f_{\mathbf{w}}) + 4\sqrt{\frac{KL\left(\mathbf{w}+\mathbf{u}\|P\right) + \ln\frac{6m}{\delta}}{m-1}}.$$

In the above expression the KL is evaluated for a fixed $\mathbf{w}$ and only $\mathbf{u}$ is random, i.e. the distribution of $\mathbf{w} + \mathbf{u}$ is the distribution of $\mathbf{u}$ shifted by $\mathbf{w}$. The lemma is analogous to similar analysis of Langford & Shawe-Taylor (2003) and McAllester (2003) obtaining PAC-Bayes margin bounds for linear predictors, and the proof, presented in Section 4, is essentially the same. As we state the lemma, it is not specific to linear separators, nor neural networks, and holds generally for any real-valued predictor.

We next show how to utilize the above general PAC-Bayes bound to prove generalization guarantees for feedforward networks based on the spectral norm of its layers.

## 2 GENERALIZATION BOUND

In this section we present our generalization bound for feedfoward networks with ReLU activations, derived using the PAC-Bayesian framework. Langford & Caruana (2001), and more recently Dziugaite & Roy (2017) and Neyshabur et al. (2017), used PAC-Bayes bounds to analyze generalization behavior in neural networks, evaluating the KL-divergence, "perturbation error" $L[f_{\mathbf{w}+\mathbf{u}}] - L[f_{\mathbf{w}}]$, or the entire bound numerically. Here, we use the PAC-Bayes framework as a tool to analytically derive a margin-based bound in terms of norms of the weights. As we saw in Lemma 1, the key to doing so is bounding the change in the output of the network when the weights are perturbed. In the following lemma, we bound this change in terms of the spectral norm of the layers:

**Lemma 2** (Perturbation Bound). *For any $B, d > 0$, let $f_{\mathbf{w}} : \mathcal{X}_{B,n} \to \mathbb{R}^k$ be a $d$-layer neural network with ReLU activations. Then for any $\mathbf{w}$, and $\mathbf{x} \in \mathcal{X}_{B,n}$, and any perturbation $\mathbf{u} = vec\left(\{U_i\}_{i=1}^d\right)$ such that $\|U_i\|_2 \leq \frac{1}{d}\|W_i\|_2$, the change in the output of the network can be bounded as follows:*

$$|f_{\mathbf{w}+\mathbf{u}}(\mathbf{x}) - f_{\mathbf{w}}(\mathbf{x})|_2 \leq eB\left(\prod_{i=1}^d \|W_i\|_2\right)\sum_{i=1}^d \frac{\|U_i\|_2}{\|W_i\|_2}.$$

This lemma characterizes the change in the output of a network with respect to perturbation of its weights, thereby bounding the sharpness of the network as defined in Keskar et al. (2016).

The proof of this lemma is presented in Section 4. Next we use the above perturbation bound and the PAC-Bayes result (Lemma 1) to derive the following generalization guarantee.

**Theorem 1** (Generalization Bound). *For any $B, d, h > 0$, let $f_\mathbf{w} : \mathcal{X}_{B,n} \to \mathbb{R}^k$ be a d-layer feedforward network with ReLU activations. Then, for any $\delta, \gamma > 0$, with probability $\geq 1 - \delta$ over a training set of size $m$, for any $\mathbf{w}$, we have:*

$$L_0(f_\mathbf{w}) \leq \widehat{L}_\gamma(f_\mathbf{w}) + \mathcal{O}\left(\sqrt{\frac{B^2 d^2 h \ln(dh)\Pi_{i=1}^d \|W_i\|_2^2 \sum_{i=1}^d \frac{\|W_i\|_F^2}{\|W_i\|_2^2} + \ln\frac{dm}{\delta}}{\gamma^2 m}}\right).$$

*Proof.* The proof involves mainly two steps. In the first step we calculate what is the maximum allowed perturbation of parameters to satisfy a given margin condition $\gamma$, using Lemma 2. In the second step we calculate the KL term in the PAC-Bayes bound in Lemma 1, for this value of the perturbation.

Let $\beta = \left(\prod_{i=1}^d \|W_i\|_2\right)^{1/d}$ and consider a network with the normalized weights $\widetilde{W}_i = \frac{\beta}{\|W_i\|_2} W_i$. Due to the homogeneity of the ReLU, we have that for feedforward networks with ReLU activations $f_{\widetilde{\mathbf{w}}} = f_\mathbf{w}$, and so the (empirical and expected) loss (including margin loss) is the same for $\mathbf{w}$ and $\widetilde{\mathbf{w}}$. We can also verify that $\left(\prod_{i=1}^d \|W_i\|_2\right) = \left(\prod_{i=1}^d \left\|\widetilde{W}_i\right\|_2\right)$ and $\frac{\|W_i\|_F}{\|W_i\|_2} = \frac{\|\tilde{W}_i\|_F}{\|\tilde{W}_i\|_2}$, and so the excess error in the Theorem statement is also invariant to this transformation. It is therefore sufficient to prove the Theorem only for the normalized weights $\tilde{\mathbf{w}}$, and hence we assume w.l.o.g. that the spectral norm is equal across layers, i.e. for any layer $i$, $\|W_i\|_2 = \beta$.

Choose the distribution of the prior $P$ to be $\mathcal{N}(0, \sigma^2 I)$, and consider the random perturbation $\mathbf{u} \sim \mathcal{N}(0, \sigma^2 I)$, with the same $\sigma$, which we will set later according to $\beta$. More precisely, since the prior cannot depend on the learned predictor $\mathbf{w}$ or its norm, we will set $\sigma$ based on an approximation $\tilde{\beta}$. For each value of $\tilde{\beta}$ on a pre-determined grid, we will compute the PAC-Bayes bound, establishing the generalization guarantee for all $\mathbf{w}$ for which $|\beta - \tilde{\beta}| \leq \frac{1}{d}\beta$, and ensuring that each relevant value of $\beta$ is covered by some $\tilde{\beta}$ on the grid. We will then take a union bound over all $\tilde{\beta}$ on the grid. For now, we will consider a fixed $\tilde{\beta}$ and the $\mathbf{w}$ for which $|\beta - \tilde{\beta}| \leq \frac{1}{d}\beta$, and hence[1] $\frac{1}{e}\beta^{d-1} \leq \tilde{\beta}^{d-1} \leq e\beta^{d-1}$.

Since $\mathbf{u} \sim \mathcal{N}(0, \sigma^2 I)$, we get the following bound for the spectral norm of $U_i$ (Tropp, 2012):

$$\mathbb{P}_{U_i \sim N(0, \sigma^2 I)}\left[\|U_i\|_2 > t\right] \leq 2he^{-t^2/2h\sigma^2}.$$

Taking a union bond over the layers, we get that, with probability $\geq \frac{1}{2}$, the spectral norm of the perturbation $U_i$ in each layer is bounded by $\sigma\sqrt{2h\ln(4dh)}$. Plugging this spectral norm bound into Lemma 2 we have that with probability at least $\frac{1}{2}$,

$$\max_{\mathbf{x} \in \mathcal{X}_{B,n}} |f_{\mathbf{w}+\mathbf{u}}(\mathbf{x}) - f_\mathbf{w}(\mathbf{x})|_2 \leq eB\beta^d \sum_i \frac{\|U_i\|_2}{\beta}$$

$$= eB\beta^{d-1} \sum_i \|U_i\|_2 \leq e^2 dB\tilde{\beta}^{d-1}\sigma\sqrt{2h\ln(4dh)} \leq \frac{\gamma}{4}, \quad (3)$$

where we choose $\sigma = \frac{\gamma}{42dB\tilde{\beta}^{d-1}\sqrt{h\ln(4hd)}}$ to get the last inequality. Hence, the perturbation $\mathbf{u}$ with the above value of $\sigma$ satisfies the assumptions of the Lemma 1.

We now calculate the KL-term in Lemma 1 with the chosen distributions for $P$ and $\mathbf{u}$, for the above value of $\sigma$.

$$KL(\mathbf{w}+\mathbf{u}\|P) \leq \frac{|\mathbf{w}|^2}{2\sigma^2} = \frac{42^2 d^2 B^2 \tilde{\beta}^{2d-2} h \ln(4hd)}{2\gamma^2} \sum_{i=1}^d \|W_i\|_F^2 \leq \mathcal{O}\left(B^2 d^2 h \ln(dh) \frac{\beta^{2d}}{\gamma^2} \sum_{i=1}^d \frac{\|W_i\|_F^2}{\beta^2}\right)$$

$$\leq \mathcal{O}\left(B^2 d^2 h \ln(dh) \frac{\Pi_{i=1}^d \|W_i\|_2^2}{\gamma^2} \sum_{i=1}^d \frac{\|W_i\|_F^2}{\|W_i\|_2^2}\right).$$

---

[1] $\left(1 + \frac{1}{d}\right)^{d-1} \leq \left(1 + \frac{1}{d}\right)^d \leq e$, as $1 + x \leq e^x$, for all $x$. Similarly $\left(1 + \frac{1}{d-1}\right)^{d-1} \leq e$, gives $\frac{1}{e} \leq \left(1 - \frac{1}{d}\right)^{d-1}$.

Hence, for any $\tilde{\beta}$, with probability $\geq 1 - \delta$ and for all $\mathbf{w}$ such that, $|\beta - \tilde{\beta}| \leq \frac{1}{d}\beta$, we have:

$$L_0(f_{\mathbf{w}}) \leq \widehat{L}_\gamma(f_{\mathbf{w}}) + \mathcal{O}\left(\sqrt{\frac{B^2 d^2 h \ln(dh) \Pi_{i=1}^d \|W_i\|_2^2 \sum_{i=1}^d \frac{\|W_i\|_F^2}{\|W_i\|_2^2} + \ln \frac{m}{\delta}}{\gamma^2 m}}\right). \tag{4}$$

Finally we need to take a union bound over different choices of $\tilde{\beta}$. Let us see how many choices of $\tilde{\beta}$ we need to ensure we always have $\tilde{\beta}$ in the grid s.t. $|\tilde{\beta} - \beta| \leq \frac{1}{d}\beta$. We only need to consider values of $\beta$ in the range $\left(\frac{\gamma}{2B}\right)^{1/d} \leq \beta \leq \left(\frac{\gamma\sqrt{m}}{2B}\right)^{1/d}$. For $\beta$ outside this range the theorem statement holds trivially: Recall that the LHS of the theorem statement, $L_0(f_{\mathbf{w}})$ is always bounded by 1. If $\beta^d < \frac{\gamma}{2B}$, then for any $\mathbf{x}$, $|f_{\mathbf{w}}(\mathbf{x})| \leq \beta^d B \leq \gamma/2$ and therefore $\widehat{L}_\gamma = 1$. Alternately, if $\beta^d > \frac{\gamma\sqrt{m}}{2B}$, then the second term in equation 2 is greater than one. Hence, we only need to consider values of $\beta$ in the range discussed above. $|\tilde{\beta} - \beta| \leq \frac{1}{d}\left(\frac{\gamma}{2B}\right)^{1/d}$ is a sufficient condition to satisfy the required condition that $|\tilde{\beta} - \beta| \leq \frac{1}{d}\beta$ in the above range, thus we can use a cover of size $dm^{\frac{1}{2d}}$. Taking a union bound over the choices of $\tilde{\beta}$ in this cover and using the bound in equation (4) gives us the theorem statement. $\qquad\square$

## 3   COMPARISON TO EXISTING GENERALIZATION BOUNDS

In this section we will compare the bound of Theorem 1 with a similar spectral norm based margin bound recently obtained by Bartlett et al. (2017a;b), as well as examine whether and when these bounds can improve over VC-based generalization guarantees.

The VC-dimension of fully connected feedforward neural networks with ReLU activation with $d$ layers and $h$ units per layer is $\tilde{\Theta}(d^2h^2)$ (Harvey et al. (2017)), yielding a generalization guarantee of the form:

$$L_0(f_{\mathbf{w}}) \leq \hat{L}_0(f_{\mathbf{w}}) + \tilde{\mathcal{O}}\left(\sqrt{\frac{d^2h^2}{m}}\right) \tag{5}$$

where here and throughout this section we ignore logarithmic factors that depend on - the failure probability $\delta$, the sample size $m$, the depth $d$ and the number of units $h$.

Bartlett et al. (2017a) showed a generalization bound for neural networks based on the spectral norm of its layers, using a different proof approach based on covering number arguments. For feedforward depth-$d$ networks with ReLU activations, and when inputs are in $\mathcal{X}_{B,n}$, i.e. are of norm bounded by $|\mathbf{x}|_2 \leq B$, their generalization guarantee, ignoring logarithmic factors, ensures that, with high probability, for any $\mathbf{w}$,

$$L_0(f_{\mathbf{w}}) \leq \widehat{L}_\gamma(f_{\mathbf{w}}) + \tilde{\mathcal{O}}\left(\sqrt{\frac{B^2 \Pi_{i=1}^d \|W_i\|_2^2 \left(\sum_{i=1}^d \left(\frac{\|W_i\|_1}{\|W_i\|_2}\right)^{2/3}\right)^3}{\gamma^2 m}}\right). \tag{6}$$

Comparing our Theorem 1 and the Bartlett et al. (2017a) bound (6), the factor $\mathcal{O}\left(\frac{1}{\gamma^2} B^2 \Pi_{i=1}^d \|W_i\|_2^2\right)$ appears in both bounds. The main difference is in the multiplicative factors, $d^2 h \sum_{i=1}^d \|W_i\|_F^2 / \|W_i\|_2^2$ in Theorem 1 compared to $\left(\sum_{i=1}^d \left(\|W_i\|_1 / \|W_i\|_2\right)^{2/3}\right)^3$ in (6). To get a sense of how these two bounds compare, we will consider the case where the norms of the weight matrices are uniform across layers—this is a reasonable situation as we already saw that the bounds are invariant to re-balancing the norm between the layers. But for the sake of comparison, we further assume that not only is the spectral norm equal across layers (this we can assume w.l.o.g.) but also the Frobenius $\|W_i\|_F$ and element-wise $\ell_1$ norm $\|W_i\|_1$ are uniform across layers (we acknowledge that this setting is somewhat favorable to our bound compared to (6)). In this case the numerator in the generalization

bound of Theorem 1 scales as:

$$O\left(d^3 h \frac{\|W_i\|_F^2}{\|W_i\|_2^2}\right),$$

(7)

while numerator in (6) scales as:

$$O\left(d^3 \frac{\|W_i\|_1^2}{\|W_i\|_2^2}\right).$$

(8)

Comparing between the bounds thus boils down to comparing $\sqrt{h}\|W_i\|_F$ with $\|W_i\|_1$. Recalling that $W_i$ is at most a $h \times h$ matrix, we have that $\|W_i\|_F \le \|W_i\|_1 \le h\|W_i\|_F$. When the weights are fairly dense and are of uniform magnitude, the second inequality will be tight, and we will have $\sqrt{h}\|W_i\|_F \ll \|W_i\|_1$, and Theorem 1 will dominate. When the weights are sparse with roughly a constant number of significant weights per unit (i.e. weight matrix with sparsity $\Theta(h)$), the bounds will be similar. Bartlett et al. (2017a) bound will dominate when the weights are extremely sparse, with much fewer significant weights than units, i.e. when most units do not have any incoming or outgoing weights of significant magnitude.

It is also insightful to ask in what regime each bound could potentially improve over the VC-bound (5) and thus provide a non-trivial guarantee. To this end, we consider the most "optimistic" scenario where $\frac{B^2}{\gamma^2}\Pi_{i=1}^d\|W_i\|_2^2 = \Theta(1)$ (it certainly cannot be lower than one if we have a non-trivial margin loss). As before, we also take the norms of the weight matrices to be uniform across layers, yielding the multiplicative factors in (7) and (8), which we must compare to the VC-dimension $\tilde{\Theta}(d^2h^2)$. We get that the bound of Theorem 1 is smaller than the VC bound if

$$\|W_i\|_F = o\left(\sqrt{h/d}\|W_i\|_2\right).$$

(9)

We always have $\|W_i\|_F \le \sqrt{h}\|W_i\|_2$, and this is tight only for orthogonal matrices, where all eigenvalues are equal. Satisfying (9), and thus having Theorem 1 potentially improving over the VC-bound, thus only requires fairly mild eigenvalue concentration (i.e. having multiple units be similar to each other), reduced rank or row-level sparsity in the weight matrices. Note that we cannot expect to improve over the VC-bound for unstructured "random" weight matrices—we can only expect norm-based guarantees to improve over the VC bound if there is some specific degenerate structure in the weights, and as we indeed see is the case here.

A similar comparison with Bartlett et al. (2017a) bound (6) and its multiplicative factor (8), yields the following condition for improving over the VC bound:

$$\|W_i\|_1 = o\left((h/\sqrt{d})\|W_i\|_2\right).$$

(10)

Since $\|W_i\|_1$ can be as large as $h^{1.5}\|W_i\|_2$, in some sense more structure is required here in order to satisfy (10), such as elementwise sparsity combined with low-rank row structure. As discussed above, Theorem 1 and Bartlett et al. (2017a) bound can each be better in different regimes. Also in terms of comparison to the VC-bound, it is possible for either one to improve over the VC bound while the other doesn't (i.e. for either (9) or (10) to be satisfied without the other one being satisfied), depending on the sparsity structure in the weights.

After the initial preprint of this paper, Bartlett et al. (2017b) presented an improved bound replacing the $\ell_1$ norm term in the bound Bartlett et al. (2017a) with $\ell_{2,1}$ norm (sum of $\ell_2$ norms of each unit). This new bound depending on $\|W_i\|_{2,1}$ is always better than the bound based on $\|W_i\|_1$ and our bound based on $\sqrt{h}\|W_i\|_F$. They match when each hidden unit has the same norm, making $\|W_i\|_{2,1} \approx \sqrt{h}\|W_i\|_F$.

## 4 PROOFS OF LEMMAS

In this section we present the proofs of Lemmas 1 and 2.

***Proof of Lemma 1.*** Let $\mathbf{w}' = \mathbf{w} + \mathbf{u}$. Let $\mathcal{S}_{\mathbf{w}}$ be the set of perturbations with the following property:

$$\mathcal{S}_{\mathbf{w}} \subseteq \left\{\mathbf{w}' \,\middle|\, \max_{\mathbf{x}\in\mathcal{X}_{B,n}} |f_{\mathbf{w}'}(\mathbf{x}) - f_{\mathbf{w}}(\mathbf{x})|_\infty < \frac{\gamma}{4}\right\}.$$

Let $q$ be the probability density function over the parameters $\mathbf{w}'$. We construct a new distribution $\tilde{Q}$ over predictors $f_{\tilde{\mathbf{w}}}$ where $\tilde{\mathbf{w}}$ is restricted to $\mathcal{S}_{\mathbf{w}}$ with the probability density function:

$$\tilde{q}(\tilde{\mathbf{w}}) = \frac{1}{Z} \begin{cases} q(\tilde{\mathbf{w}}) & \tilde{\mathbf{w}} \in \mathcal{S}_{\mathbf{w}} \\ 0 & \text{otherwise.} \end{cases}$$

Here $Z$ is a normalizing constant and by the lemma assumption $Z = \mathbb{P}\left[\mathbf{w}' \in \mathcal{S}_{\mathbf{w}}\right] \geq \frac{1}{2}$. By the definition of $\tilde{Q}$, we have:

$$\max_{\mathbf{x} \in \mathcal{X}_{B,n}} |f_{\tilde{\mathbf{w}}}(\mathbf{x}) - f_{\mathbf{w}}(\mathbf{x})|_{\infty} < \frac{\gamma}{4}.$$

Therefore, the perturbation can change the margin between two output units of $f_{\mathbf{w}}$ by at most $\frac{\gamma}{2}$; i.e. for any perturbed parameters $\tilde{\mathbf{w}}$ drawn from $\tilde{Q}$:

$$\max_{i,j \in [k], \mathbf{x} \in \mathcal{X}_{B,n}} |(|f_{\tilde{\mathbf{w}}}(\mathbf{x})[i] - f_{\tilde{\mathbf{w}}}(\mathbf{x})[j]|) - (|f_{\mathbf{w}}(\mathbf{x})[i] - f_{\mathbf{w}}(\mathbf{x})[j]|)| < \frac{\gamma}{2}$$

Since the above bound holds for any $\mathbf{x}$ in the domain $\mathcal{X}_{B,n}$, we can get the following a.s.:

$$L_0(f_{\mathbf{w}}) \leq L_{\frac{\gamma}{2}}(f_{\tilde{\mathbf{w}}})$$
$$\widehat{L}_{\frac{\gamma}{2}}(f_{\tilde{\mathbf{w}}}) \leq \widehat{L}_{\gamma}(f_{\mathbf{w}})$$

Now using the above inequalities together with the equation (2), with probability $1 - \delta$ over the training set we have:

$$L_0(f_{\mathbf{w}}) \leq \mathbb{E}_{\tilde{\mathbf{w}}}\left[L_{\frac{\gamma}{2}}(f_{\tilde{\mathbf{w}}})\right]$$

$$\leq \mathbb{E}_{\tilde{\mathbf{w}}}\left[\widehat{L}_{\frac{\gamma}{2}}(f_{\tilde{\mathbf{w}}})\right] + 2\sqrt{\frac{2(KL\left(\tilde{\mathbf{w}} \| P\right) + \ln \frac{2m}{\delta})}{m-1}}$$

$$\leq \widehat{L}_{\gamma}(f_{\mathbf{w}}) + 2\sqrt{\frac{2(KL\left(\tilde{\mathbf{w}} \| P\right) + \ln \frac{2m}{\delta})}{m-1}}$$

$$\leq \widehat{L}_{\gamma}(f_{\mathbf{w}}) + 4\sqrt{\frac{KL\left(\mathbf{w}' \| P\right) + \ln \frac{6m}{\delta}}{m-1}},$$

The last inequality follows from the following calculation.

Let $\mathcal{S}_{\mathbf{w}}^c$ denote the complement set of $\mathcal{S}_{\mathbf{w}}$ and $\tilde{q}^c$ denote the density function $q$ restricted to $\mathcal{S}_{\mathbf{w}}^c$ and normalized. Then,

$$KL(q\|p) = ZKL(\tilde{q}\|p) + (1-Z)KL(\tilde{q}^c\|p) - H(Z),$$

where $H(Z) = -Z \ln Z - (1-Z)\ln(1-Z) \leq 1$ is the binary entropy function. Since KL is always positive, we get,

$$KL(\tilde{q}\|p) = \frac{1}{Z}\left[KL(q\|p) + H(Z)) - (1-Z)KL(\tilde{q}^c\|p)\right] \leq 2(KL(q\|p)+1). \qquad \square$$

***Proof of Lemma 2***. Let $\Delta_i = \left|f_{\mathbf{w}+\mathbf{u}}^i(\mathbf{x}) - f_{\mathbf{w}}^i(\mathbf{x})\right|_2$. We will prove using induction that for any $i \geq 0$:

$$\Delta_i \leq \left(1 + \frac{1}{d}\right)^i \left(\prod_{j=1}^i \|W_j\|_2\right) |\mathbf{x}|_2 \sum_{j=1}^i \frac{\|U_j\|_2}{\|W_j\|_2}.$$

The above inequality together with $\left(1 + \frac{1}{d}\right)^d \leq e$ proves the lemma statement. The induction base clearly holds since $\Delta_0 = |\mathbf{x} - \mathbf{x}|_2 = 0$. For any $i \geq 1$, we have the following:

$$\begin{aligned}
\Delta_{i+1} &= \left|(W_{i+1} + U_{i+1})\,\phi_i(f_{\mathbf{w}+\mathbf{u}}^i(\mathbf{x})) - W_{i+1}\phi_i(f_{\mathbf{w}}^i(\mathbf{x}))\right|_2 \\
&= \left|(W_{i+1} + U_{i+1})\left(\phi_i(f_{\mathbf{w}+\mathbf{u}}^i(\mathbf{x})) - \phi_i(f_{\mathbf{w}}^i(\mathbf{x}))\right) + U_{i+1}\phi_i(f_{\mathbf{w}}^i(\mathbf{x}))\right|_2 \\
&\leq (\|W_{i+1}\|_2 + \|U_{i+1}\|_2)\left|\phi_i(f_{\mathbf{w}+\mathbf{u}}^i(\mathbf{x})) - \phi_i(f_{\mathbf{w}}^i(\mathbf{x}))\right|_2 + \|U_{i+1}\|_2\left|\phi_i(f_{\mathbf{w}}^i(\mathbf{x}))\right|_2 \\
&\leq (\|W_{i+1}\|_2 + \|U_{i+1}\|_2)\left|f_{\mathbf{w}+\mathbf{u}}^i(\mathbf{x}) - f_{\mathbf{w}}^i(\mathbf{x})\right|_2 + \|U_{i+1}\|_2\left|f_{\mathbf{w}}^i(\mathbf{x})\right|_2 \\
&= \Delta_i(\|W_{i+1}\|_2 + \|U_{i+1}\|_2) + \|U_{i+1}\|_2\left|f_{\mathbf{w}}^i(\mathbf{x})\right|_2,
\end{aligned}$$

where the last inequality is by the Lipschitz property of the activation function and using $\phi(0) = 0$. The $\ell_2$ norm of outputs of layer $i$ is bounded by $|\mathbf{x}|_2 \Pi_{j=1}^i \|W_j\|_2$ and by the lemma assumption we have $\|U_{i+1}\|_2 \leq \frac{1}{d} \|W_{i+1}\|_2$. Therefore, using the induction step, we get the following bound:

$$\Delta_{i+1} \leq \Delta_i \left(1 + \frac{1}{d}\right) \|W_{i+1}\|_2 + \|U_{i+1}\|_2 |\mathbf{x}|_2 \prod_{j=1}^i \|W_j\|_2$$

$$\leq \left(1 + \frac{1}{d}\right)^{i+1} \left(\prod_{j=1}^{i+1} \|W_j\|_2\right) |\mathbf{x}|_2 \sum_{j=1}^i \frac{\|U_j\|_2}{\|W_j\|_2} + \frac{\|U_{i+1}\|_2}{\|W_{i+1}\|_2} |\mathbf{x}|_2 \prod_{j=1}^{i+1} \|W_i\|_2$$

$$\leq \left(1 + \frac{1}{d}\right)^{i+1} \left(\prod_{j=1}^{i+1} \|W_j\|_2\right) |\mathbf{x}|_2 \sum_{j=1}^{i+1} \frac{\|U_j\|_2}{\|W_j\|_2}. \qquad \square$$

## 5 Conclusion

In this paper, we presented new perturbation bounds for neural networks thereby giving a bound on its sharpness. We also discussed how PAC-Bayes framework can be used to derive generalization bounds based on the sharpness of a model class. Applying this to the feedforward networks, we showed that a tighter generalization bound can be achieved based on the spectral norm and Frobenius norm of the layers. The simplicity of the proof compared to that of covering number arguments in Bartlett et al. (2017a) suggest that the PAC-Bayes framework might be an important tool in analyzing the generalization behavior of neural networks.

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
