# OpenReview forum: "A PAC-Bayesian Approach to Spectrally-Normalized Margin Bounds for Neural Networks"
_ICLR.cc/2018/Conference — Accept (Poster)_

### Official Review · AnonReviewer1 · 2017-11-09
**Elegant**

**Rating:** 9
**Confidence:** 4

**Review:**

The authors prove a generalization guarantee for deep
neural networks with ReLU activations, in terms of margins of the
classifications and norms of the weight matrices.  They compare this
bound with a similar recent bound proved by Bartlett, et al.  While,
strictly speaking, the bounds are incomparable in strength, the
authors of the submission make a convincing case that their new bound
makes stronger guarantees under some interesting conditions.

The analysis is elegant.  It uses some existing tools, but brings them
to bear in an important new context, with substantive new ideas needed.
The mathematical writing is excellent.

Very nice paper.

I guess that networks including convolutional layers are covered by
their analysis.  It feels to me that these tend to be sparse, but that
their analysis still my provides some additional leverage for such
layers.  Some explicit discussion of convolutional layers may be
helpful.

---

### Official Review · AnonReviewer2 · 2017-11-27
**A PAC-Bayesian generalization bound**

**Rating:** 6
**Confidence:** 3

**Review:**

This paper provides a new generalization bound for feed forward networks based on a PAC-Bayesian analysis. The generalization bound depends on the spectral norm of the layers and the Frobenius norm of the weights. The resulting generalization bound is similar (though not comparable) to a recent result of Bartlett et al (2017), however the technique is different since this submission uses PAC-Bayesian analysis. The resulting proof is more simple and streamlined compared to that of Bartlett et al (2017).

The paper is well presented, the result is explained and compared to other results, and the proofs seem correct. The result is not particularly different from previous ones, but the different proof technique might be a good enough reason to accept this paper.




Typos: Several citations are unparenthesized when they should be. Also, after equation (6) there is a reference command that is not compiled properly.

---

### Official Review · AnonReviewer3 · 2017-11-30
**A simple proof for another generalization bounds on ReLU NNs**

**Rating:** 7
**Confidence:** 4

**Review:**

This paper combines a simple PAC-Bayes argument with a simple perturbation analysis (Lemma 2) to get a margin based generalization error bound for ReLU neural networks (Theorem 1) which depends on the product of the spectral norms of the layer parameters as well as their Frobenius norm. The main contribution of the paper is the simple proof technique to derive Theorem 1, much simpler than the one use in the very interesting work [Bartlett et al. 2017] (appearing at NIPS 2017) which got an analogous bound but with a dependence on the l1-norm of the layers instead of the Frobenius norm. The authors make a useful comparison between these bounds in Section 3 showing that none is dominating the others, but still analyzing their properties in terms of structural properties of the weight matrices.

I enjoyed reading this paper. One could think that it makes a somewhat incremental contribution with respect to the more complete work (both theory and practice) from [Bartlett et al. 2017]. Nevertheless, the simplicity and elegance of the proof as well as the result might be useful for the community to get progress on the theoretical analysis of NNs.

The paper is well written, though I make some suggestions for the camera ready version below to improve clarity.

I verified most of the math.

== Detailed suggestions ==

1) The authors should specify in the abstract and in the introduction that they are analyzing feedforward neural networks *with ReLU activation functions* so that the current context of the result is more transparent. It is quite unclear how one could generalize the Theorem 1 to arbitrary activation functions phi given the crucial use of the homogeneity of the ReLU at the beginning of p.4. Though the proof of Lemma 2 only appears to be using the 1-Lipschitzness property of phi as well as phi(0) =0. (Unless they can generalize further; I also suggest that they explicitly state in the (interesting) Lemma 2 that it is for the ReLU activations (like they did in Theorem 1)).

2) A footnote (or citation) could be useful to give a hint on how the inequality 1/e beta^(d-1) <= tilde{beta}^(d-1) <= e beta^(d-1) is proven from the property |beta-tilde{beta}|<= 1/d beta (middle of p.4).

3) Equation (3) -- put the missing 2 subscript for the l2 norm of |f_(w+u)(x) - f_w(x)|_2 on the LHS (for clarity).

4) One extra line of derivation would be helpful for the reader to rederive the bound|w|^2/2sigma^2  <= O(...) just above equation (4). I.e. first doing the expansion keeping the beta terms and Frobenius norm sum, and then going directly to the current O(...) term.

5) bottom of p.4: use hat{L}_gamma = 1 instead of L_gamma =1 for more clarity.

6) Top of p.5: the sentence "Since we need tilde{beta} to satisfy (...)" is currently awkwardly stated. I suggest instead to say that "|tilde{beta}- beta| <= 1/d (gamma/2B)^(1/d) is a sufficient condition to have the needed condition |tilde{beta}-beta| <= 1/d beta over this range, thus we can use a cover of size dm^(1/2d)."

7) Typo below (6): citetbarlett2017...

8) Last paragraph p.5: "Recalling that W_i is *at most* a hxh matrix" (as your result do not require constant size layers and covers the rectangular case).

---

### Decision · Program_Chairs · 2018-01-29
**ICLR 2018 Conference Acceptance Decision**

**Decision:**

Accept (Poster)

**Comment:**

This is a strong paper presenting a very clean proof of a result that is similar, though now incomparable to one due to Bartlett et al. These bounds (and Bartlett's) are among the most promising norm-based bounds for NNs.

I would simply add that the citation of Dziugaite and Roy (2017) could be improved. There work also connects sharpness (or flatness) with generalization via the PAC-Bayes framework, and moreover, there bounds are nonvacuous.  Are the bounds in this paper nonvacuous, say, on MNIST for 60,000 training data, for the network learned by SGD?  If not, how close do they get to 1.0?